# Aerobic exercise and action observation priming modulate functional connectivity

**Jasper I. Mark, Hannah Ryan, Katie Fabian, Kaitlin DeMarco, Michael D. Lewek[ID], Jessica M. Cassidy[ID]***

Department of Health Sciences, University of North Carolina at Chapel Hill, Chapel Hill, North Carolina, United States of America

* jcassidy@med.unc.edu

**Data Availability Statement:** All de-identified participant files are available from UNC's Dataverse open-source platform: (https://doi.org/10.15139/S3/0CWTZC).

## Abstract

Aerobic exercise and action observation are two clinic-ready modes of neural priming that have the potential to enhance subsequent motor learning. Prior work using transcranial magnetic stimulation to assess priming effects have shown changes in corticospinal excitability involving intra- and interhemispheric circuitry. The objective of this study was to determine outcomes exclusive to priming- how aerobic exercise and action observation priming influence functional connectivity within a sensorimotor neural network using electroencephalography. We hypothesized that both action observation and aerobic exercise priming would alter resting-state coherence measures between dominant primary motor cortex and motor-related areas in alpha (7–12 Hz) and beta (13–30 Hz) frequency bands with effects most apparent in the high beta (20–30 Hz) band. Nine unimpaired individuals (24.8 ± 3 years) completed a repeated-measures cross-over study where they received a single five-minute bout of action observation or moderate-intensity aerobic exercise priming in random order with a one-week washout period. Serial resting-state electroencephalography recordings acquired from 0 to 30 minutes following aerobic and action observation priming revealed increased alpha and beta coherence between leads overlying dominant primary motor cortex and supplementary motor area relative to pre- and immediate post-priming timepoints. Aerobic exercise priming also resulted in enhanced high beta coherence between leads overlying dominant primary motor and parietal cortices. These findings indicate that a brief bout of aerobic- or action observation-based priming modulates functional connectivity with effects most pronounced with aerobic priming. The gradual increases in coherence observed over a 10 to 30-minute post-priming window may guide the pairing of aerobic- or action observation-based priming with subsequent training to optimize learning-related outcomes.

## Introduction

Priming is a form of implicit (nonconscious) memory whereby prior experience or exposure to a stimulus shapes current behavior [1]. While priming originated from the cognitive psychology field, its fundamental concepts have infused rehabilitation research [2, 3]. The

**Funding:** The author(s) received no specific funding for this work.

**Competing interests:** The authors have declared that no competing interests exist.

transient modifications in synaptic function imparted by priming have the potential to heighten the effects of a subsequent goal-mediated event occurring during motor rehabilitation training. Indeed, priming the primary motor cortex (M1) and related cortical regions with the delivery of non-invasive neurostimulation [4–7] pharmacology [8], movement [9, 10], and mental imagery [11], have been investigated as strategies to enhance learning-related neuroplasticity and clinical rehabilitation outcomes. Priming may therefore serve a valuable role in neurorehabilitation when paired with a subsequent intervention such as physical training.

Aerobic exercise and action observation are two clinic-ready modes of priming neural connections. Past work employing a single bout of aerobic exercise-based priming in non-clinical cohorts demonstrated enhanced motor skill acquisition [12] and motor learning outcomes [13, 14] along with heightened responses to neuromodulation [13, 15, 16]. Action observation, a type of motor imagery that involves watching an individual perform a goal-directed activity [2], uses the mirror neuron system [17] to engage brain regions typically active during movement execution. These areas include M1, premotor, and parietal cortices and supplementary motor area [18–21]. Akin to aerobic exercise priming, action observation priming has also enhanced the effects of subsequent motor training [22–24] and neuromodulation [21] and, thus, may be a viable alternative in the clinical setting for those unable to safely participate in aerobic exercise. Despite these encouraging findings, the underlying mechanisms of priming are not fully understood. Elucidating these mechanisms will enrich our understanding of how priming readies the neural system for motor skill acquisition and learning, which may therefore optimize learning-related outcomes associated with the pairing of priming and the training/intervention.

Discussions on the purported mechanisms of aerobic exercise and action observation priming [25, 26] have focused on the neurophysiological effects such as the release of brain-derived neurotrophic factor (BDNF) [13] and increased corticospinal excitability, often involving M1, as demonstrated by transcranial magnetic stimulation (TMS) studies [27–30] and cerebral blood flow [31, 32]. Others have reported alterations in hippocampal long-term potentiation [33], neurogenesis [34], and protein structure [35] along with changes in BDNF-mediated synaptic transmission [36] from aerobic exercise training. Grey matter volume [37] and cortical activation [38] changes have also been reported from action observation training.

A single aerobic exercise session also modulated various intracortical and interhemispheric circuits [39]. These particular findings emphasize the growing interest and examination of functional neural network connectivity to characterize brain states, assess intervention efficacy, and predict one's learning and/or rehabilitation outcomes [40]. Determining the effects of priming on functional connectivity may supplement past work evaluating neural excitability (i.e., TMS) by providing a richer account of how priming augments neural circuits and networks. The examination of priming-induced alterations in functional connectivity may reinforce the concept of shared neuroplasticity mechanisms (akin to those described above) between aerobic exercise and action observation but may also reveal distinct mechanisms. Knowledge pertaining to the engagement of both mutual and distinct functional connections with these priming approaches may foster novel circuit-specific therapies or priming strategies to enhance learning and/or rehabilitation effects.

Rather than examining outcomes arising from the pairing of priming with an intervention or training, the focus here was to examine outcomes exclusive to priming. The purpose of this work was to therefore determine how aerobic- and action observation-based priming augment functional neural connectivity as determined by changes in neural oscillatory coherence using electroencephalography (EEG). The attributes of EEG, including its portability, quick set-up and application, and capability of directly capturing neural activity, make it an ideal modality

for studying the effects of priming. We hypothesized that both action observation and aerobic exercise priming would alter resting-state EEG measures of coherence between dominant M1 and motor-related regions in alpha (7–12 Hz) and beta (13–30 Hz) frequency bands, which are bands associated with visuospatial attention [41–43] and motor function [44], respectively. Based on past EEG findings that demonstrated changes in functional connectivity in the beta frequency band following an acute bout of exercise [45], we further hypothesized that the effects of aerobic exercise priming on EEG coherence would favor the beta band, specifically the high beta (20–30 Hz) frequency range. Likewise, while EEG studies examining action observation have shown changes in cortical oscillations involving both alpha [46] and beta [47] bands, EEG work specifically examining coherence demonstrated alterations in the alpha band [48]. We therefore hypothesized that changes in coherence following action observation priming would be most apparent in the alpha frequency band.

## Materials and methods

### Participants

We recruited participants between the ages of 18 and 30 years old from the University of North Carolina at Chapel Hill. Additional inclusion criteria included right handedness (to minimize variability due to discrepancies in hemisphere dominance across participants) as confirmed by the Edinburgh Handedness Inventory, no history of cardiovascular or neurologic diagnoses, sufficient cognitive function as assessed by the Montreal Cognitive Assessment, and the ability to tolerate five minutes of moderate aerobic activity. Individuals were excluded from study participation if they had a resting blood pressure ≥ 180/110 mmHg. All participants provided written informed consent as approved by the Institutional Review Board at the University of North Carolina at Chapel Hill.

### Procedures

**Study design.** Participants completed one baseline visit and two priming visits as part of a repeated measures cross-over study with a one-week washout period between priming visits (Fig 1). During the baseline visit, participants completed a medical history questionnaire and a behavioral battery evaluating mood (Beck Depression Inventory), executive function (Trails Making Test B), physical activity level (General Practice Physical Activity Questionnaire, GPPAQ), balance (Mini Balance Evaluation Systems Test, Mini BESTest), and self-efficacy (Activities-specific Balance Confidence (ABC) Scale). Investigators also examined upper- and lower-extremity motor function using the Nine-Hole Peg Test and gait speed. Participants' gait speed was used to establish parameters associated with the aerobic priming condition. We included the above assessments of mood, self-efficacy, executive function, dexterity, etc. because these factors may either influence one's response to priming [49, 50] or outcomes associated with motor learning [51, 52] when these priming modes are paired with a physical training intervention, which is the intent of future work. During this visit, participants were also filmed while walking on a treadmill at a self-selected speed. Video recordings involved one-minute clips of anterior, lateral, and posterior views of the individual. Video footage was later incorporated in the action observation priming condition (described below). In preparation for the priming visits, participants were instructed to avoid caffeine consumption and exercise within one hour of the visit. Participants were also encouraged to maintain a consistent medication regimen as necessary throughout study participation. At the start of each priming visit, investigators collected participants' resting heart rate (HR) and blood pressure. Participants completed a series of 3-minute resting-state EEG recordings occurring before, immediately after, and at 10, 20, and 30 minutes following priming.

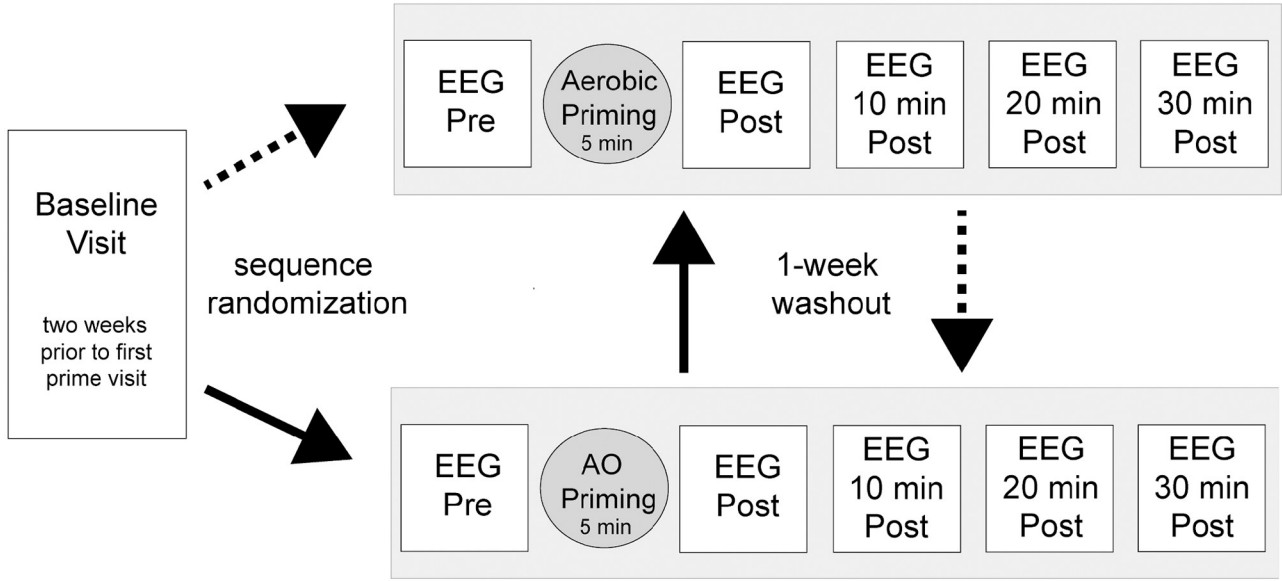

**Fig 1. Experiment and study design.** Participants completed one baseline visit and two priming visits (grey boxes) separated by a one-week washout period. Priming order was randomized to either aerobic exercise priming first and action observation priming second (dotted arrows) or action observation priming first followed by aerobic exercise priming a week later (solid arrows). During each priming visit, participants completed five three-minute resting-state electroencephalography recordings before, after, and at 10, 20, and 30 minutes following a 5-minute bout of priming.

**Priming.** Participants were randomized to one of two priming sequences: aerobic exercise at the first priming visit and then action observation at the second priming visit OR action observation priming at the first priming visit and then aerobic priming at the second priming visit (Fig 1). A one-week washout period was used between priming conditions to minimize the potential influence of one priming condition on the other. Each priming condition lasted approximately five minutes.

The aerobic priming condition involved participants walking on a treadmill maintaining their HR in a targeted zone based on maximum HR, computed as 208 –(0.7 * age), and resting HR [53]. The lower limit of the target HR zone was calculated as (0.5 * (maximum HR—resting HR) + resting HR), and the upper limit of the target HR zone was (0.7 * (maximum HR—resting HR) + resting HR) [54]. Investigators monitored participants' HR and oxygen saturation during the priming bout at one-minute intervals. Participants were instructed to only increase treadmill speed in order to maintain their targeted HR zone.

The action observation priming condition involved participants viewing a five-minute video on a 22-inch computer monitor positioned approximately 3-feet from seated participants. The video entailed a montage of 15-second clips of unimpaired individuals walking on a treadmill at a self-selected pace. The video also included five randomly inserted clips of the participant walking on the treadmill (previously recorded at the baseline visit). To ensure consistent attention throughout the intervention, participants were instructed beforehand to count the number of clips they viewed of themselves walking.

**EEG acquisition and pre-processing.** We acquired five three-minute resting-state EEG recordings with a dense-array 256-lead Hydrocel net (Electrogeodesics Inc., Eugene, OR) at each visit. During collection, participants were instructed to remain still while sitting upright with support. Throughout the recording, participants focused on a central fixation cross projected on a laptop computer screen positioned approximately two feet from them. Data were

sampled at 1,000 Hz using a high input impedance Net Amp 400 amplifier and Net Station 4.5 software (Electrogeodesics Inc.).

We transferred the raw and unfiltered EEG data to Matlab (R2017b, MathWorks, Natick, MA) for offline preprocessing using EEGLAB [55] version 2019.0. Data processing involved: re-referencing to the average signal across all leads after the removal of 64 electrodes overlying cheek and neck regions, 40 Hz low-pass filtering, 0.5 Hz high-pass filtering, segmentation into one-second non-overlapping epochs, visual inspection for muscle artifact removal, and ocular and cardiac artifact removal using an Infomax independent components analysis [55], and then one last visual inspection to remove any remaining artifact. Pre-processed EEG then underwent a surface Laplacian transformation to mitigate volume conduction effects [56].

**EEG coherence measurements.** Using a fast Fourier transform, we calculated measures of coherence in frequency bands the alpha (7–12 Hz), low beta (13–19 Hz), and high beta (20–30 Hz) frequency bands. These particular frequency bands contribute to cognitive and motor function [41–44]. The primary seed region for coherence measurements involved a set of predefined electrodes (C3 and the surrounding six leads) overlying left M1 (lM1, dominant hemisphere) [57, 58]. The C3 lead corresponds to the precentral gyrus, and previous work has shown that EEG activity recorded from this lead reflects M1 activity [59]. We examined interhemispheric functional connectivity between electrodes overlying lM1 with an additional set of predefined electrodes overlying rM1 (C4 and surrounding six electrodes) and also intrahemispheric functional connectivity between lM1 and predefined leads overlying supplementary motor area (SMA) and left parietal (lPr) and dorsal premotor (lPMd) cortical regions. Several of these regions were included in prior work examining performance-related gains on a visuomotor task following visuospatial training [60]. A list of all predefined regions with corresponding electrode numbers is provided in the S1 Table. Coherence measurements were computed as the squared correlation coefficient. Values ranged from 0 to 1 with values approaching 1 signifying consistency of phase and amplitude ratios across time in a given frequency.

**Statistical analysis.** Statistical analyses were performed in JMP Pro 16.0.0 (SAS Inc., Cary, NC). This study involved a two-period (week one, week two), two-condition (aerobic exercise priming, action observation priming) repeated-measures crossover design with a one-week washout period. We measured coherence during each visit at five timepoints: pre, post, and 10, 20, and 30 minutes post-priming (post10, post20, and post30). To assess differences in coherence across timepoints (within-condition change) and between priming conditions, we employed a mixed-effects linear model. Fixed effects were *priming condition* and *timepoint*. Each participant served as a random intercept to model within-subject correlation.

To ensure that significant changes in coherence did not arise from the effects of carryover, period, and priming sequence, we performed several screening procedures. We assessed pre timepoints for significant carryover and period effects by running the model with *priming condition*, *period*, and *sequence* as fixed effects. We also screened for unequal carryover, period effects, and interactions between priming condition and period by running the main model with *priming condition*, *period*, and *sequence* as fixed effects. Post-hoc analyses involved pairwise comparisons using Tukey's honestly significant difference with an alpha level of .05 denoting significance. Model assumptions of normal distribution of residuals and homoskedasticity were confirmed with Shapiro-Wilk and Levene tests, respectively.

Though we did not conduct a formal power analysis given the pilot nature of this work, we refer to recent and related work depicting significant changes in EEG activity following action observation priming in both unimpaired [61] and clinical [50] cohorts. In the former, a small sample of young adults (n = 6) ranging in age from 18–27 years completed approximately 4 minutes of action observation priming (viewing left/right elbow flexion) delivered in 6-second

increments followed by a motor imagery task of the same action [61]. Investigators observed significant gains in event-related desynchronization (diminished EEG power; -33.50% to -36.09%) in leads overlying lM1 and rM1 compared to the control priming condition (viewing arrows indicative of left/right elbow flexion, -22.82% to -23.70%) [61]. Similar findings were also observed in a small (n = 11) heterogeneous cohort of individuals with stroke ranging in age from 37–76 years and 18–1919 days post-stroke with 6 participants depicting aphasia, apraxia, frontal lobe dysfunction, and/or unilateral spatial neglect [50]. Participants completed approximately 3 minutes of action observation-based priming over the course of the experiment, and investigators observed a significant enhancement of event-related desynchronization in a similar frequency band (30 ± 5.0%) as compared to the motor imagery condition (12.2 ± 3.9%) [50]. Despite discrepancies in EEG outcome measures (coherence vs. event-related desynchronization), the similarities between these studies [50, 61] and ours including sample size, abbreviated priming bouts, and utilization of EEG to assess priming, bolster the reliability of this work, which is fundamental to the future implementation of short priming sessions in clinical settings.

## Results

Nine individuals (7 females, average age ± standard deviation: 24.8 ± 3 years) completed study procedures. Overall, the cohort demonstrated normal balance (Mini BESTest: 27.8 ± 0.44) and high balance self-efficacy (ABC Scale: 96.1 ± 2.7%). Participants primarily reported themselves as *active* (n = 6) with the remaining as *moderately active* (n = 2) and *inactive* (n = 1) as defined by the GPPAQ. Table 1 provides additional cohort detail.

Participants reached their target heart rate in 178.8 ± 74 seconds after they started walking on the treadmill when the priming condition commenced. The average time elapsed between the ending of the aerobic exercise priming condition to the first post-prime EEG recording was 199.1 ± 71 seconds. During the action observation priming condition, all participants correctly reported the number of video clips that featured themselves walking on the treadmill. The average time elapsed between the end of the action observation priming condition to the first post-prime EEG recording was 46.0 ± 28 seconds. Continuous wear of the EEG cap during the experiment necessitated additional time following the aerobic priming condition to check placement and impedance of EEG leads to ensure sufficient signal quality for post-prime EEG recordings.

**Table 1. Participant demographics.**

| Subject | Sex | Age (yrs) | Priming Sequence | BDI normal = 0 | TMT-B (sec) | 9-HPT (D/ND) | Gait speed (m/sec) | RHR (bpm) | THRR (bpm) |
|---|---|---|---|---|---|---|---|---|---|
| 1 | F | 24 | A-AO | 0 | 58.2 | 0.82 | 1.28 | 70 | 135–155 |
| 2 | M | 30 | A-AO | 0 | 48.5 | 0.75 | 1.27 | 52 | 120–147 |
| 3 | M | 24 | A-AO | 5 | 53.4 | 1.22 | 1.31 | 86 | 139–160 |
| 4 | F | 20 | A-AO | 6 | 52.7 | 0.91 | 1.58 | 90 | 142–163 |
| 5 | F | 32 | AO-A | 5 | 56.2 | 0.98 | 1.31 | 75 | 130–152 |
| 6 | F | 24 | A-AO | 0 | 31.1 | 0.97 | 1.49 | 78 | 135–158 |
| 7 | F | 23 | AO-A | 2 | 70.6 | 1.02 | 1.59 | 61 | 126–153 |
| 8 | F | 22 | A-AO | 3 | 60.0 | 1.06 | 1.23 | 73 | 133–157 |
| 9 | F | 24 | AO-A | 6 | 49.9 | 0.96 | 1.38 | 65 | 128–153 |

A-AO, aerobic exercise priming before action observation priming; AO-A, action observation priming before aerobic exercise priming; BDI, Beck Depression Inventory; D/ND, ratio of dominant to non-dominant hand performance; F, female; M, male; RHR, resting heart rate; THRR, Target Heart Rate Range; TMT-B, Trail Making Test B

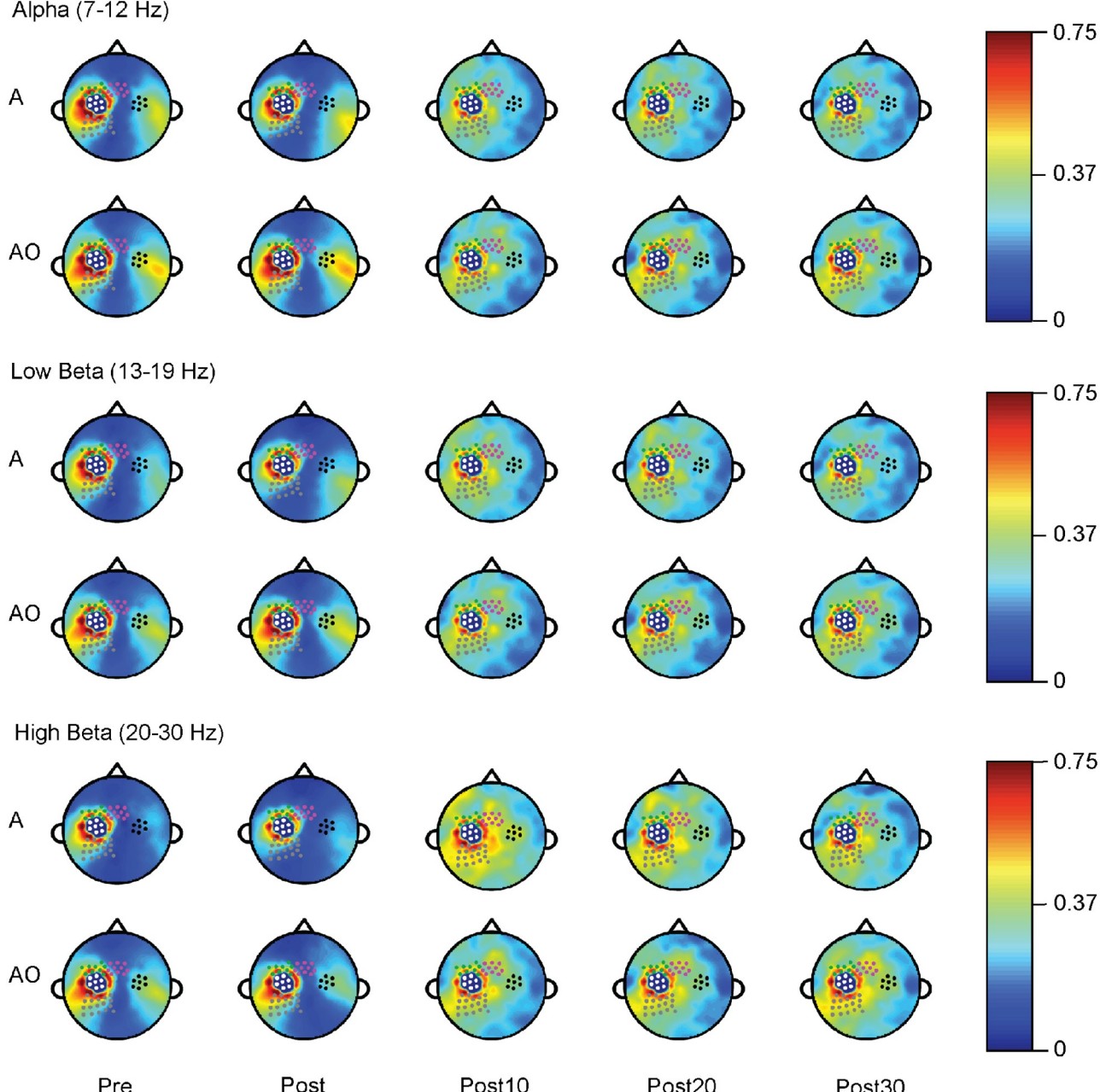

**Fig 2. EEG coherence pre/post priming.** Aerobic exercise (A) and action observation (AO) priming resulted in enhanced EEG coherence in alpha (7–12 Hz, top), low beta (13–19 Hz, middle), and high beta (20–30 Hz, bottom) frequency bands most pronounced at 10, 20, and 30 minutes post-priming (Post10, Post20, and Post30). White leads represent the coherence seed region overlying left primary motor cortex. Black leads denote right primary motor cortex, grey leads denote left parietal cortex, green leads denote dorsal premotor cortex, and magenta leads denote supplementary motor area.

Both aerobic and action observation priming conditions enhanced alpha, low beta, and high beta EEG coherence in a predefined sensorimotor network (Fig 2). As hypothesized, modulations of EEG coherence following aerobic exercise priming were most abundant in the high beta frequency band. The results summarized below represent significant findings following the screening of pre timepoints and the main model to assess for undesirable carryover, period effects, and interactions between condition and period. Figs 3–6 illustrate individual-

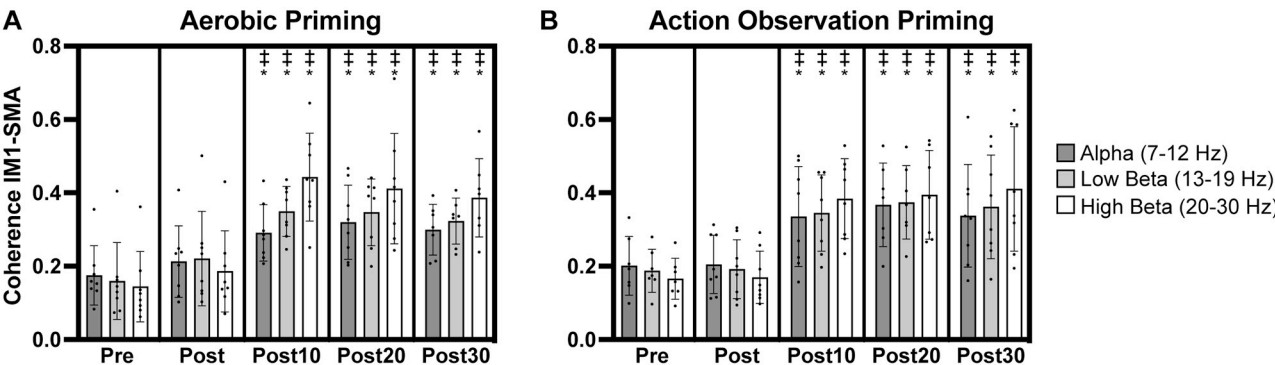

**Fig 3. Coherence in leads overlying left primary motor (lM1) cortex and supplementary motor area (SMA).** Significant increases in alpha, low beta, and high beta lM1-SMA coherence occurred after aerobic (A) and action observation (B) priming. Individual participant data points illustrated along with group averages and standard deviations. ‡ indicates significant increase from pre timepoint. * indicates significant increase from post timepoint.

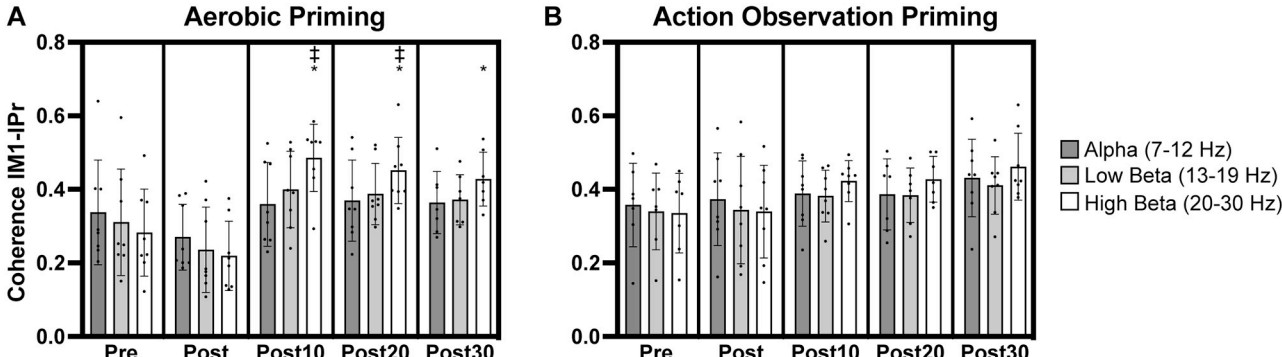

**Fig 4. Coherence in leads overlying left primary motor (lM1) and left parietal (lPr) cortices.** Increases in high beta lM1-lPr coherence occurred after aerobic priming (A). ‡ indicates significant change from pre timepoint. * indicates significant increase from post timepoint.

and group-level coherence measurements between leads overlying lM1 and SMA (Fig 3), lPr (Fig 4), rM1 (Fig 5), and lPMd (Fig 6) for aerobic exercise and action observation priming across all timepoints and frequencies. Additional group-level coherence information is provided in the S2–S4 Tables.

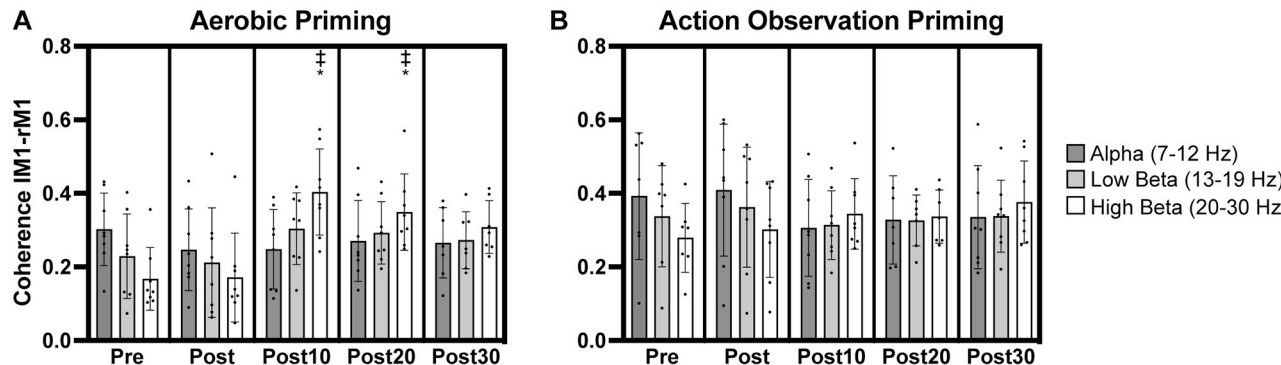

**Fig 5. Coherence in leads overlying left primary motor (lM1) and right primary motor (rM1) cortices.** Increases in high beta lM1-rM1 coherence occurred following aerobic priming (A). ‡ indicates significant increase from pre timepoint. * indicates significant increase from post timepoint.

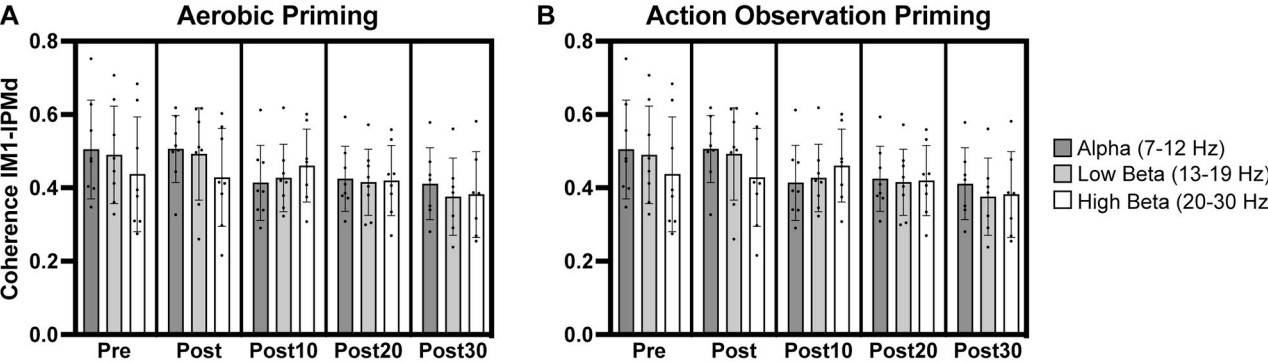

**Fig 6. Coherence in leads overlying left primary motor (lM1) and left dorsal premotor (lPMd) cortices.** Aerobic (A) and action observation (B) priming did not result in any significant change in alpha, low beta, or high beta lM1-lPMd coherence.

## Coherence: lM1-SMA

For alpha coherence between leads overlying lM1 and SMA (lM1-SMA), a significant effect of *timepoint* was observed ($F_{(4,59.2)} = 11.49$, p < .0001) with post-hoc analyses revealing significantly greater coherence at post10, post20, and post30 timepoints compared to pre (t = 4.36–5.25, p = .0001-.0005) and also at post10, post20, and post30 timepoints compared to post (t = 3.64–4.55, p = .0003-.005) (Fig 3).

Similar findings for low and high beta lM1-SMA coherence were also observed. For low beta lM1-SMA coherence, there was a significant effect of *timepoint* ($F_{(4,59.4)} = 18.43$, p < .0001) with post-hoc analyses indicating significant increases in coherence at post10, post20, and post30 timepoints relative to pre (t = 5.75–6.32, p < .0001) and post (t = 4.62–5.18, p < .0001-.0002, Fig 3) timepoints. For high beta lM1-SMA coherence, a significant effect of *timepoint* occurred ($F_{(4,59.3)} = 32.7$, p < .0001) with post-hoc analyses again indicating significantly greater coherence at post10, post20, and post30 timepoints relative to pre (t = 7.46–8.07, p < .0001) and post (t = 6.79–7.40, p < .0001) timepoints (Fig 3).

## Coherence: lM1-lPr

A significant interaction between *priming condition* and *timepoint* occurred ($F_{(4,59.0)} = 3.07$, p = .023) for high beta coherence between leads overlying lM1 and lPr (lM1-lPr). Post-hoc analyses showed significantly greater high beta lM1-lPr coherence at post10 and post20 timepoints relative to the pre timepoint for the aerobic priming condition (t = 4.19–5.04, p = .0002-.003, Fig 4A). Significantly greater high beta lM1-lPr coherence at post10, post20, and post30 timepoints compared to the immediate post timepoint for the aerobic priming condition also occurred (t = 4.80–6.62, p < .0001-.0004, Fig 4A).

## Coherence: lM1-rM1

A similar interaction between *priming condition* and *timepoint* resulted for high beta coherence between leads overlying bilateral M1 cortices (lM1-rM1, $F_{(4,59.9)} = 3.78$, p = .008). Post-hoc analyses demonstrated significantly greater coherence at post10 and post20 timepoints compared to pre (t = 4.39–5.71, p < .0001-.002) and post (t = 4.30–5.62, p < .0001-.002) timepoints following the aerobic priming condition (Fig 5A). The non-normal distribution of residuals involving high beta lM1-rM1 coherence (W = 0.968, p = .05), however, necessitates caution with the interpretation of these findings.

### Coherence: lM1-lPMd

We did not observe significant modulations in alpha, low beta, and high beta coherence in leads overlying left M1 and lPMd (lM1-lPMd) following aerobic and action observation priming (Fig 6).

## Discussion

The purpose of this study was to determine how two clinic-ready modes of motor system priming influence functional connectivity as measured by EEG coherence. Our preliminary findings indicate that both aerobic and action observation priming resulted in increased alpha and beta lM1-SMA coherence. Aerobic priming also resulted in increased high beta lM1-lPr and lM1-rM1 coherence. Expanding on previous TMS work showing alterations in corticospinal excitability following similar modes of priming, this work demonstrates that a relatively short (five-minute) bout of priming also mediates neural oscillatory coherence likely reflecting alterations in neural communication [62] that may foster a favorable neural environment for subsequent learning.

Informed by past research associating attention/visuospatial [41–43] and motor system [44] function with alpha and beta bands, we chose to focus on these frequencies in our current work. We observed increases in alpha and beta lM1-SMA coherence with both action observation and aerobic exercise priming and also increased high beta lM1-lPr coherence following aerobic exercise priming. These findings likely reflect both the attentional and sensorimotor processing involved in action observation and aerobic exercise. Prior work has demonstrated interactions between attention and motor system function, including alterations in SMA activity [63], and also the involvement of parietal cortex in somatosensory and cognitive processes, including selective attention [64]. Additionally, more recent fMRI work examining changes in cortical activity following aerobic exercise [65] and action observation [20] have shown modulations in postcentral, secondary somatosensory, and parietal regions which reinforce our identified functional connections from EEG.

Notably, neither mode of priming significantly impacted coherence between leads overlying lM1 and lPMd. In contrast to motor training, motor priming is not goal-directed nor does it entail skill-based training [25]. Several potential explanations may account for this finding. A lack of modulation of lM1-lPMd coherence emphasizes these training vs. priming distinctions. Given the role of the premotor cortex in motor planning, this finding may also reflect minimal motor planning requirements associated with aerobic exercise and action observation priming. These findings parallel prior action observation work. In a similar study examining action observation of gait in two cohorts of subacute stroke (n = 5) and unimpaired controls (n = 9), investigators observed increased activation in inferior parietal, frontal, and temporal cortical regions in both cohorts [20]. Further, a meta-analysis of 139 neuroimaging studies to determine cortical areas associated with action observation reported a bilateral neural network involving similar regions [66] that are consistent with regions comprising the mirror neuron system [17]. While the focus of this work was directed at the sensorimotor system, examination of functional connections within the mirror neuron system following motor priming may be important in subsequent motor training outcomes. This finding may also reveal universal challenges associated with EEG, particularly dense-array EEG, where leads in close proximity capture similar signals thus resulting in elevated coherence.

We recognize that our coherence finding of enhanced high beta lM1-lPr coherence following aerobic priming depicts specificity with regards to both frequency and priming, which echoes findings from previous EEG work demonstrating distinct mu (8–12 Hz) and beta (18–25 Hz) band EEG topographies within a sensorimotor network during motor imagery and motor

execution [67]. Also, Dal Maso et al. [45] observed increases in alpha and beta band functional connectivity following an acute bout of aerobic exercise with greater abundance of exercise-induced effects in the beta band. Because there was no ensuing motor training session and retention test included in our study, it is difficult to surmise how priming and frequency-specific changes in coherence impact learning outcomes, or at least the improvement in motor performance following a single training session. However, previous findings of high beta coherence between leads overlying lM1 and lPr areas significantly predicted greater motor skill acquisition in a group of young (18–30 years), unimpaired participants [68], which underscores the importance of this functional connection and frequency band that only aerobic exercise priming elicited. Additional work is necessary to examine how priming-induced changes in functional connectivity contribute to the enhancement of subsequent motor learning. Nevertheless, our findings indicate that aerobic exercise priming results in more pronounced changes in coherence in a frequency band central to motor functioning, which may equate to an enhanced environment for future motor training and learning.

Another important observation from this work was that priming-induced changes in coherence took at least 10 minutes to develop and persisted for at least 30 minutes. These findings align with other work that also featured multiple post-test sessions, including a session immediately after aerobic exercise completion [39]. Potential factors mediating this timeframe may relate to changes in cerebral blood flow or the release of various neurotrophic factors, such as BDNF and vascular endothelial growth factor along with the release of catecholamines such as norepinephrine and dopamine [26]. Inconsistencies in the literature pertaining to the upregulation of BDNF following acute bouts of exercise [13, 15], for example, encourage further investigation of these factors including their possible role in action observation priming. Importantly, our findings of increased coherence occurring over 10 to 30 minutes post-priming also delineates a temporal window that corresponds to an optimal neural environment for potential learning. Commencing motor training during this timeframe or structuring a training session whereby training-related demands on the individual peak during this window may result in better learning-related outcomes. Further, our findings of no pre to immediate post differences in coherence suggests that a brief "pause" between priming and subsequent motor training may be warranted to capitalize on the priming.

A major discrepancy between our work and that of others examining aerobic exercise and action observation-based priming was duration. Participants in our study completed a considerably shorter bout of priming in comparison to the 20 to 30-minute bouts featured in other studies [13, 15, 22, 27, 39, 65, 69]. With the ultimate goal of clinical implementation, specifically in stroke rehabilitation, we assert that a five-minute bout of priming possesses greater clinical feasibility given that the average treatment session lasts only 30 minutes. To our knowledge, there exists no other short duration aerobic training study. In addition to those studies of short duration action observation training referenced above [50, 61], work by Hioka et al. [20] involved participants viewing six 30-second blocks of gait observation alternating with 6 30-second blocks of rest for a total of 6 minutes in length. Comparison of observation and rest conditions by investigators revealed activation in regions consistent with the mirror neuron system in the observation condition. We therefore recognize the need for additional work to establish and confirm the internal validity of our priming protocol.

Relatedly, the intensity of aerobic exercise priming in this work overlaps with others [15, 39, 65, 69] and adheres to clinical recommendations [70]. The use of a target heart rate range not derived from maximal exercise testing (i.e., VO2 max) further ensures efficient clinical translation. The low-to-moderate intensity range employed here may also lessen exercise-induced cortisol [71], which has been shown to negatively impact neuroplasticity [72] and future learning [73]. Confirming if similar priming-induced changes in functional connectivity occur in a

stroke population is a next step, but it requires judicious consideration of other personal factors including prior level of function, stroke severity, comorbidities, cognitive level, fitness experience, time since stroke, and medication use [74]. Furthermore, the post-stroke neural landscape, typified by immense neuroplasticity potential and shifts in glutamatergic and GABAergic transmission [75], presents several future research directions in the context of sensorimotor priming. In addition, exercise intensity appears to regulate resting-state neural connectivity in networks supporting attention and cognitive processing, sensorimotor function, affect, and reward [76]. Modulations in coherence observed here may roughly correspond to an "in-between" exercise intensity state as participants exercised at a low to moderate intensity. Future work employing a similar experimental design but with higher intensity activity may observe discrepancies in the up or downregulation of coherence and also differences in the onset and lasting effects of coherence modulation as compared to findings observed in this work. The consideration of exercise intensity in an aerobic-based priming study is therefore imperative and likely requires adjustment when accounting for disease progression and/or the timeframe of recovery following neural injury such as stroke.

## Strengths and limitations

This study contains several notable strengths including the examination of two clinically feasible modes of priming, use of a cross-over study design, and the application of EEG as a tool to determine priming effects on functional connectivity. Serial EEG coherence measurements is another strength of this study. Previous work examining within-session reliability of resting-state EEG coherence measurements in 40 healthy young adults confirmed the reliability of both alpha and beta coherence measurements (r = .80-.97), which supports changes in coherence due to priming vs. random variability [77].

We also acknowledge limitations with the current work. In line with the pilot nature of this study, our sample size was small. We encourage caution with the interpretation of our findings, including the interaction between *timepoint* and *condition* interaction involving leads overlying lM1-lPr for aerobic exercise priming as differences in high beta coherence between conditions and/or large standard deviation values at pre timepoints may have contributed to the finding. Additionally, our sample was comprised primarily of physically active females, which limits the generalizability of our findings. Additional work to validate these findings, assess potential covariates (e.g., physical activity level, age, mood, etc.), and confirm the feasibility and efficacy of these priming modes in a clinical cohort are warranted. We also recognize that six of the nine participants were randomized to the A-AO sequence; however, the non-significant effect of *sequence* when added to the model mitigated this issue. Though we attempted to standardize priming duration between aerobic exercise and action observation priming, participants typically required an additional three minutes of treadmill walking to achieve their target heart rate range. The average time from the completion of priming to the start of EEG recording was also generally longer for the aerobic exercise condition. It is unknown how a slight delay in acquisition time of the immediate post-priming EEG recording influenced our overall findings. Given the increase in coherence beginning at 10-minutes post-priming, we suspect that these delays had only a minimal effect.

Though much of the neuroimaging literature in aerobic exercise and action observation has focused primarily on the activity of cortical regions, activity modulation from subcortical structures may also occur [65], which highlights the inherent limitation of EEG as confined to the cortex. We acknowledge that our predefined electrode groups, including the lM1 seed region, were based on average location, as source localization (other than a spatial Laplacian transform) and head modeling did not occur. Also, our EEG analyses involved the

examination of coherence between a seed region (dominant, lM1) and predefined groups of electrodes overlying secondary motor regions. The possibility exists that aerobic exercise and/ or action observation priming augments functional connections outside of the sensorimotor neural network. Indeed, prior work has shown aerobic exercise-induced alterations in cognitive neural networks involving dorsolateral prefrontal cortex [78], for example. Therefore, examination of whole brain connectivity and the incorporation of data-driven approaches as done previously [58] may reveal other relevant connections and networks. The global changes in coherence (Fig 2) observed following action observation may also arise not only from the *action* ingredient (walking) but also from ingredients related to selective and sustained attention. For instance, in our action observation priming condition, 75% of the video clips presented to participants entailed other people walking, and participants were asked beforehand to count the number of clips of themselves walking. As action observation involves the viewing of purposeful activity to eventually mimic or replicate during practice, a similar priming approach in a stroke rehabilitation setting might entail participants viewing typical gait patterns from unimpaired individuals for the majority of the priming session with occasional clips of themselves walking depicting atypical gait. As previous work has shown differences in brain activity evoked from different perspectives (e.g., first- vs. third-person) [79], the coherence findings presented here may differ according to attentional features of the action observation condition. Future work should discern specific contributions of action vs. attention on coherence to determine if these ingredients synergistically enhance subsequent motor learning.

## Conclusions

Priming represents an efficient and potentially cost-effective strategy to enhance the effects of subsequent goal-mediated training. Motivated by past work demonstrating transient modifications in corticospinal excitability following aerobic exercise- [14, 27, 39] and action observation-based [28, 29] priming, this study sought to determine how these two clinic-ready modes of priming modulated functional connectivity. Our main finding was that serial resting-state EEG recordings collected before and up to 30 minutes following priming showed increases in alpha, low beta, and high beta coherence between leads overlying dominant M1 (seed region) and other motor-related regions, including SMA and lPr cortices. These functional connections tended to strengthen over time. Collectively, this work compliments past TMS work examining intra- and intercortical circuitry by providing preliminary findings that enrich our understanding of the underlying mechanisms exclusive to priming. The findings presented here may also inform future studies of clinically-feasible dosing (priming intensity and duration) and temporal pairing with subsequent motor training. Confirming the presence (or absence) of similar connectivity findings in a clinical cohort such as stroke is an essential next step.

## Supporting information

**S1 Table. Electrode numbers for predefined regions.**
(DOCX)

**S2 Table. Alpha (7–12 Hz) coherence.**
(DOCX)

**S3 Table. Low beta (13–19 Hz) coherence.**
(DOCX)

**S4 Table. High beta (20–30 Hz) coherence.**
(DOCX)

## Acknowledgments

The authors would like to thank Dr. Wanqing Zhang, MD, PhD and Ramis Chowdhury, Hrishika Muthukrishnan, Etienne Jeangilles, Eric Zheng, and Sara Galante for their assistance on this project.

## Author Contributions

**Conceptualization:** Michael D. Lewek, Jessica M. Cassidy.

**Data curation:** Jasper I. Mark, Hannah Ryan, Katie Fabian.

**Formal analysis:** Jessica M. Cassidy.

**Investigation:** Jasper I. Mark, Hannah Ryan, Katie Fabian, Kaitlin DeMarco, Jessica M. Cassidy.

**Methodology:** Michael D. Lewek, Jessica M. Cassidy.

**Project administration:** Jessica M. Cassidy.

**Resources:** Michael D. Lewek, Jessica M. Cassidy.

**Supervision:** Michael D. Lewek.

**Visualization:** Jasper I. Mark, Jessica M. Cassidy.

**Writing – original draft:** Jessica M. Cassidy.

**Writing – review & editing:** Jasper I. Mark, Hannah Ryan, Katie Fabian, Kaitlin DeMarco, Michael D. Lewek.

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
