## [Decision Letter · Decision Letter 0]

17 Jan 2023

PONE-D-22-27025Aerobic exercise and action observation priming modulate functional connectivityPLOS ONE

Dear Dr. Cassidy,

Thank you for submitting your manuscript to PLOS ONE. After careful consideration, we feel that it has merit but does not fully meet PLOS ONE’s publication criteria as it currently stands. Therefore, we invite you to submit a revised version of the manuscript that addresses all the points raised during the review process as outlined herein. As you can read in the assessment completed by 2 independent reviewers, which also corroborated my own reading, your manuscript does not meet point 3 of PLOS ONE's publication criteria (analyses conducted and reporting of analyses). It is crucial that you take into account all points the reviewers made regarding your analyses and reporting of these. Note that both authors highlight the small participant sample. It is important to address their comments to reach publishable standard with a small sample size of 9 participants for your design. On that note, please do provide more information on the G-Power analysis conducted and the link to other studies as suggested by reviewer 2. I also strongly agree with their request to provide the currently missing rationales for the technical and methodological choices made. Note that reviewer 1 also made an important comment regarding the relevance of your findings (point 3) which I think will strengthen the theoretical discussion of your findings.  In addition to the reviewers' comments; please also address the role of 'priming' in your manuscript. Your study is on the effect of a motor related activity (aerobic exercise or action observation) on resting state functional connectivity, but you are setting it within the framework of 'priming'. The rationale for this needs to be clearer. Please explain in the manuscript why you are using the term 'priming' when you have not included a task that either preceded or followed the treatment. Similarly, the rationale for the choice of the two interventions is limited. Please provide a stronger rationale for including both conditions in the same study protocol. Finally, you refer to the unbalanced order due to the randomization. Why have you not taken this into account as a factor in the model? This could have addressed this issue. 

We look forward to receiving your revised manuscript.

Kind regards,

Corinne Jola

Academic Editor

PLOS ONE

Journal Requirements:

Reviewers' comments:

Reviewer's Responses to Questions

**Comments to the Author**

1. Is the manuscript technically sound, and do the data support the conclusions?

Reviewer #1: Partly

Reviewer #2: Partly

2. Has the statistical analysis been performed appropriately and rigorously? 

Reviewer #1: I Don't Know

Reviewer #2: Yes

3. Have the authors made all data underlying the findings in their manuscript fully available?

Reviewer #1: Yes

Reviewer #2: Yes

4. Is the manuscript presented in an intelligible fashion and written in standard English?

Reviewer #1: Yes

Reviewer #2: Yes

5. Review Comments to the Author

Reviewer #1: Interesting paper which opens a potential direction for rehabilitation: movement observation. I’ve been waiting for that branch to be better studied since the rehabilitative effects of sound observation are already relatively widely explored, like music listening in stroke and Parkinson rehabilitation. Therefore, the approach of this paper is relevant and timely.

I have a few remarks for which is kindly ask you to see the file attached.

Reviewer #2: The authors presented a study investigating the priming effect of action observation and aerobic exercise on modulating functional connectivity between primary motor and motor-related areas during resting-state. They recorded EEG before and after (three times) priming and reported increased coherence in the alpha and beta frequency bands following both priming sessions. The manuscript is well-organized, the language is appropriate, and the topic is engaging. I appreciated the battery of tests used to evaluate the participants, the "strengths and limitations" section, and the goal of testing a short period of priming to be used in clinical settings. That said, I encourage the authors to consider a series of points that could strengthen their work that, in my opinion, is not ready for publication at this stage of work.

- The main point of concern regards the study's statistical power due to a quite limited number of participants and the 4x2 experimental design. The authors reported such an issue in the "Strengths and limitations" section, which I appreciated, but I'm not entirely convinced that acknowledging it solved the problem. The authors also reported a power analysis based on a previous study. However, such a study was conducted more than 25 years ago when the standards for reproducibility were different. Also, in that study, the authors recorded brain activity via a single channel and did not perform a coherence analysis between electrodes. I suggest the authors better structure this section of the manuscript, reporting, for instance, further examples of studies with similar sample sizes and discussing their results according to this point. The possibility of increasing the sample size would ideally represent a perfect solution. Given the future potential of using short priming sessions in clinical settings, the reliability of the results is essential.

- Please consider reporting more detail regarding the action observation priming (e.g., the size of the stimuli and distance from the screen).

- Please consider reporting references in support of the choice of the "predefined electrodes" used for each region of interest (e.g., C3 for M1), and report the actual names/numbers of the electrodes considered (e.g., "surrounding six leads").

- Please consider reporting references (and rationale) supporting the authors' choice throughout the study, for instance, for the use of the questionnaires/tests, frequency bands selected, and other technical/methodological decisions.

- Please consider reporting a graphic representation of the significant results (e.g., boxplots, raincloud plots) to help the reader understand the differences between conditions (e.g., pre, post, post10, etc.). It is not completely clear if the differences reported and discussed between post10, post20 and post30 were found only in the high beta band after exercise priming. A rewording of the "results" section to clarify postdoc interactions would be helpful.

6. PLOS authors have the option to publish the peer review history of their article (what does this mean?). If published, this will include your full peer review and any attached files.

Reviewer #1: **Yes: **Hanna Poikonen, PhD MSc

Reviewer #2: No

---

## [Decision Letter · Decision Letter 1]

21 Mar 2023

Aerobic exercise and action observation priming modulate functional connectivity

PONE-D-22-27025R1

Dear Dr. Cassidy,

We’re pleased to inform you that your manuscript has been judged scientifically suitable for publication and will be formally accepted for publication once it meets all outstanding technical requirements.

Please note that the additional reviewer who provided an assessment of your revised manuscript has made two helpful comments (with reference suggestions) below, i.e., action observation from post-stroke cohort/issues regarding sample size. You may want to consider including a brief statement for either of those points. 

Kind regards,

Corinne Jola

Academic Editor

PLOS ONE

Additional Editor Comments (optional):

Reviewers' comments:

Reviewer's Responses to Questions

**Comments to the Author**

1. If the authors have adequately addressed your comments raised in a previous round of review and you feel that this manuscript is now acceptable for publication, you may indicate that here to bypass the “Comments to the Author” section, enter your conflict of interest statement in the “Confidential to Editor” section, and submit your "Accept" recommendation.

Reviewer #1: All comments have been addressed

Reviewer #3: All comments have been addressed

2. Is the manuscript technically sound, and do the data support the conclusions?

Reviewer #1: Yes

Reviewer #3: Yes

3. Has the statistical analysis been performed appropriately and rigorously? 

Reviewer #1: Yes

Reviewer #3: Yes

4. Have the authors made all data underlying the findings in their manuscript fully available?

Reviewer #1: Yes

Reviewer #3: Yes

5. Is the manuscript presented in an intelligible fashion and written in standard English?

Reviewer #1: Yes

Reviewer #3: Yes

6. Review Comments to the Author

Reviewer #1: Really interesting paper, great work! All my previous remarks were discussed in a suitable manner. I hope this preliminary work will lead to similar studies with a larger sample size.

Reviewer #3: Thank you for the opportunity to read this manuscript. I have read both versions as well as both reviews. I do not have any major concerns about the paper as it is well-organized, the methods are detailed, and the results are clearly presented. My only criticism is that I did not understand why the authors recommend using action observation of gait with both normal (healthy population) and abnormal (post-stroke) examples. Upper limb action observation studies do not use both types of movement. From a clinical perspective, using impaired movement during action observation is not intuitive and may be counterproductive. Below are comments regarding the perception that the sample size is too small.

Byblow and colleagues[1] published a manuscript in this journal which included healthy participants completing five separate experiments. Three of the five experiments used a design like the study described here. They compared the neural mechanisms of bilateral symmetrical movement (a priming technique) with bilateral asymmetrical movement. The number of participants in each experiment ranged from n=6 to n=13. Significant results were found in 4 of the 5 studies. Another aerobics priming study also used a cross-over repeated measures design in athletes. There were 5 priming conditions and a total of 14 participants.[2] Power analysis was not mentioned in either study. Neural mechanistic studies that examine changes after one session of priming are needed. It is difficult to estimate power in a first-time study. The fact that there was significance between pre- and post-priming EEG measures reported here may indicate that the sample size is large enough for the research objective. The explanation of the sample size in the statistical analysis section is tedious and, perhaps could be condensed. However, the study should be published.

1. Byblow, W.D., et al., Mirror Symmetric Bimanual Movement Priming can Increase Corticomotor Excitability and Enhance Motor Learning. PLoS ONE, 2012. 7(3): p. e33882.

2. Matsumoto, T., Y. Tomita, and K. Irisawa, Identifying the Optimal Arm Priming Exercise Intensity to Improve Maximal Leg Sprint Cycling Performance. J Sports Sci Med, 2023. 22(1): p. 58-67.

7. PLOS authors have the option to publish the peer review history of their article (what does this mean?). If published, this will include your full peer review and any attached files.

Reviewer #1: **Yes: **Dr. Hanna Poikonen, Professorship for Learning Sciences and Higher Education, Department of Humanities, Social and Political Sciences, Swiss Federal Institute of Technology Zurich (ETH Zurich), Switzerland

Reviewer #3: No

---

## [Editor Report · Acceptance letter]

28 Mar 2023

PONE-D-22-27025R1 

Aerobic exercise and action observation priming modulate functional connectivity 

Dear Dr. Cassidy:

I'm pleased to inform you that your manuscript has been deemed suitable for publication in PLOS ONE. Congratulations! Your manuscript is now with our production department. 

Kind regards, 

on behalf of

Dr. Corinne Jola 

Academic Editor

PLOS ONE